# Ginsenoside Compound K Protects against Obesity through Pharmacological Targeting of Glucocorticoid Receptor to Activate Lipophagy and Lipid Metabolism

**DOI:** 10.3390/pharmaceutics14061192

**Published:** 2022-06-02

**Authors:** Siwen Yang, Ting Liu, Chenxing Hu, Weili Li, Yuhan Meng, Haiyang Li, Chengcheng Song, Congcong He, Yifa Zhou, Yuying Fan

**Affiliations:** 1Engineering Research Center of Glycoconjugates of Ministry of Education, Jilin Provincial Key Laboratory of Chemistry and Biology of Changbai Mountain Natural Drugs, School of Life Sciences, Northeast Normal University, Changchun 130024, China; yangsw776@nenu.edu.cn (S.Y.); liut700@nenu.edu.cn (T.L.); hucx099@nenu.edu.cn (C.H.); liwl506@nenu.edu.cn (W.L.); mengyh242@nenu.edu.cn (Y.M.); lihy589@nenu.edu.cn (H.L.); songcc225@nenu.edu.cn (C.S.); 2Department of Cell and Molecular Biology, Feinberg School of Medicine, Northwestern University, Chicago, IL 60611, USA; congcong.he@northwestern.edu

**Keywords:** ATGL, ginsenoside CK, GR, lipophagy, lipid metabolism

## Abstract

(1) Background: The glucocorticoid receptor (GR) plays a key role in lipid metabolism, but investigations of GR activation as a potential therapeutic approach have been hampered by a lack of selective agonists. Ginsenoside compound K (CK) is natural small molecule with a steroid-like structure that offers a variety of therapeutic benefits. Our study validates CK as a novel GR agonist for the treatment of obesity. (2) Methods: By using pulldown and RNA interference, we determined that CK binds to GR. The anti-obesity potential effects of CK were investigated in obese mice, including through whole-body energy homeostasis, glucose and insulin tolerance, and biochemical and proteomic analysis. Using chromatin immunoprecipitation, we identified GR binding sites upstream of lipase ATGL. (3) Results: We demonstrated that CK reduced the weight and blood lipids of mice more significantly than the drug Orlistat. Proteomics data showed that CK up-regulated autophagy regulatory proteins, enhanced fatty acid oxidation proteins, and decreased fatty acid synthesis proteins. CK induced lipophagy with the initial formation of the phagophore via AMPK/ULK1 activation. However, a blockade of autophagy did not disturb the increase in CK on lipase expression, suggesting that autophagy and lipase are independent pathways in the function of CK. The pulldown and siRNA experiments showed that GR is the critical target. After binding to GR, CK not only activated lipophagy, but also promoted the binding of GR to the ATGL promoter. (4) Conclusions: Our findings indicate that CK is a natural food candidate for reducing fat content and weight.

## 1. Introduction

Obesity is complex chronic disease caused by the environment, genetic factors, and an energy balance dysregulation. Obese patients exhibit white adipose accumulation in their body, resulting from the long-term storage of excess energy [1,2,3,4]. Numerous medications for obesity treatment have been developed, mainly involving reducing appetite and thereby energy intake, or increasing energy expenditure, or lowering calorie absorption [5]. The acceleration of fat breakdown is a direct and effective strategy for countering obesity. Intracellular triglycerides (TGs) exist in the form of lipid droplets (LDs). It is well known that the hydrolysis of TGs depends on multiple lipases, including adipose triglyceride lipase (ATGL), hormone-sensitive lipase (HSL), and monoacylglyceryl lipase (MGL). Recently, it was proposed that TGs could also be hydrolyzed using acid lipase present in autolysosomes [6,7,8].

Lipophagy is a type of autophagy involving the selective degradation of lipids, which promotes the transport of TGs and lipoproteins to lysosomes for degradation [9,10]. However, the mechanisms underlying this process are yet to be fully understood. After entering the autophagosome, the LD fuses with the lysosome, and lysosomal acid lipase hydrolyzes various substrates, such as TGs and cholesterol esters [11,12]. A heat-shock protein (hsp70) recognizes the pentapeptide sequence of the target protein, and transfers it to the surface of the lysosome to bind to lysosome-associated membrane protein 2A. This protein, along with hsp70, forms a complex that translocates the substrate proteins to lysosomes for degradation [13,14,15]. LDs are encapsulated by members of the perilipin family, among which PLIN2 and PLIN3 block the contact between LD and ATGL and prevent lipid breakdown [16,17].

Autophagy is a process for the degradation of cellular materials for maintaining homeostasis. Defects in autophagy are common in multiple diseases, such as cancer, neurodegenerative disorders, obesity, and diabetes [18,19]. Therefore, autophagy activators may have the potential to improve disease conditions [20,21]. In our previous study, we identified glycoconjugates as novel autophagy inducers that can regulate metabolic diseases. Among the plant-derived metabolites screened, ginsenoside Rg2 has been reported to improve insulin sensitivity, as well as learning and memory in mice with Alzheimer’s disease through autophagy induction [22].

Ginsenoside is small molecule derived from traditional herbs, including American ginseng, ginseng, and panax notoginseng. These include monomeric ginsenosides with similar structures that display abundant biological activity, such as Rg2, Rg3, and CK. Ginsenoside CK is the main metabolite and final absorption form of protopanaxadiol-type ginsenoside in the human intestine, showing anti-hyperlipidemia and anti-tumor effects [23]. Previous studies demonstrated that CK decreased the expression of leptin, which is synthesized and secreted by adipocytes, and is directly implicated in the lipogenic pathways of 3T3-L1 adipocytes [24]. However, whether this action contributes to stimulating autophagy remains to be determined. We have reported that CK-stimulated autophagy was involved in human colon cancer apoptosis [25], but the molecular mechanism of CK to activate autophagy and other pharmacological activities through autophagy are unclear. In particular, the targets for the activation of autophagy are yet to be identified.

Herein, we investigated the novel mechanisms and targets adopted by ginsenoside CK for regulating lipid metabolism. We hypothesized that autophagy-inducing ginsenoside CK may lower body weight and blood lipid. To test this hypothesis, we further studied the signaling pathways and target receptors of ginsenoside CK in obese mice.

## 2. Materials and Methods

### 2.1. Preparation of Ginsenosides

Ginsenosides CK was prepared using biotransformation as described in a previous study [26]. The transformation products were separated using an HPLC system with an analytical Shim-pack PREP-ODS (H) column (4.6 mm × 250 mm, 5 μm) connected to an HPLC system (Shimadzu, Kyoto, Japan), and eluted at a flow rate of 1.0 mL/min using the following gradient program: 0–10 min, 32% acetonitrile, 10–40 min, 32–60% acetonitrile, 40–50 min, and 60% acetonitrile, monitored by measuring the absorbance at 203 nm.

### 2.2. Cell Culture

Mouse hepatoma Hepa1-6 cells, human cervical carcinoma HeLa cells, human hepatoma HepG2 cells, and human lung carcinoma A549 cells were purchased from the ATCC. The cells were cultured in DMEM (25 mM glucose, 11995040, Gibco) supplemented with 10% fetal bovine serum (FBS), 1% penicillin-streptomycin, and 1% HEPES (BOSTER, PYG0019) at 37 °C and 5% CO_2_. GFP-LC3/HeLa cells were maintained in DMEM (25 mM glucose), and supplemented with 0.4 mg/mL G418 (10131035, Thermo Scientific, Waltham, MA, USA) at 37 °C and 5% CO_2_.

For inhibitory assays, the antagonists or inhibitors, such as RU486 (30 μM, M8046, Sigma-Aldrich, St. Louis, MO, USA), were used to treat Hepa1-6 cells for 12 h; similarly, BafA1 (100 nM, B1793, Sigma-Aldrich) and 3-MA (10 mM, M9281, Sigma-Aldrich) were used to treat GFP-LC3/HeLa cells or Hepa1-6 cells for indicated time.

### 2.3. Animals and Treatments

Male C57BL/6J (wild-type mice) and ob/ob (B6/JGpt-*Lep*^em1Cd25^/Gpt) mice were obtained from GemPharmatech Co., Ltd. (Nanjing, China). At 5 weeks of age, mice were housed in a temperature-controlled facility (21 °C, 12 h light/12 h dark cycle, 60–70% humidity). Standard laboratory food (60% cereals, 33% protein, and 3% oil) and water were given ad libitum. After 1 week of feeding, the mice were subjected to the different treatments at 6 weeks of age (25 ± 5 g). Body weight and blood glucose were measured weekly. All experiments were approved by the Animal Care and Use Committee of Northeast Normal University (SYXK 2018-0015). At the end of the experiments, serum and organ samples were collected for the determination of biochemical parameters. Further details about the biochemical analysis and histology methods are included in the Supporting Information (Appendix B).

The mice were treated with 20 mg/kg of the ginsenoside CK or DMSO (vehicle) through i.p. injection once a day for five weeks. The mice were orally administered 150 mg/kg orlistat as a positive control.

### 2.4. Whole-Body Energy Homeostasis

Oxygen consumption (VO_2_), carbon dioxide output (VCO_2_), RER, energy expenditure, and food intake were measured using a Comprehensive Laboratory Animal Monitoring System (Columbus Instruments, Columbus, OH, USA). The mice were housed individually in metabolic cages with access to food and water ad libitum. After the mice were acclimatized to the metabolic cages for 24 h, data were recorded at intervals of 6 min, and the mice were monitored for 24 h. The O_2_ and CO_2_ contents in sample air from individual cages were evaluated using sensors. VO_2_ was calculated as the difference between the input oxygen flow and the output oxygen flow. The VO_2_ and VCO_2_ values were used to calculate the RER and heat production.

### 2.5. GTT and ITT

For GTT, the mice were fasted for 6 h, followed by an intraperitoneal injection of glucose (2 g/kg body weight). For ITT, the mice were fasted for 6 h, followed by an injection of insulin (1.5 U/kg body weight). Blood glucose levels were monitored at 0, 15, 30, 60, and 120 min after injection using a One Touch Ultra Easy Glucometer (Johnson, New Brunswick, NJ, USA).

### 2.6. Western Blotting

Homogenized tissues and cells were lysed in lysis buffer containing phosphatase and protease inhibitors. The samples were centrifuged at 12,000 rpm at 4 °C for 20 min to separate the supernatant, and the protein concentration was determined using the Coomassie Brilliant Blue assay. Approximately 10 μg of the samples were dispensed into the wells, along with the Precision Plus Protein^TM^ Dual Color Standard (Bio-Rad) for SDS-PAGE, following which the samples were transferred to PVDF membranes. The membranes were blocked by treating with 5% skim milk with Tween 20 (PBST) for 1 h. Subsequently, the membranes were treated using specific primary antibodies and secondary antibodies. Further details about the methods are included in the Supporting Information (Appendix B).

### 2.7. Autophagy Analyses

GFP-LC3/HeLa cells were treated for 3 h with a normal medium, a starvation medium (Earle’s balanced salt solution; E7510, Sigma-Aldrich), and a normal medium containing 35 μM of ginsenoside CK. The cells were washed with PBS, and fixed using 4% paraformaldehyde for 15 min. Intracellular puncta were quantified using fluorescence microscopy (Olympus, Tokyo, Japan).

For the in vivo evaluation of autophagy, 5-week-old male C57BL/6J mice were administered CK (20 mg/kg) via i.p. injection once a day for three days. The autophagy inhibitor chloroquine (C6628, Sigma-Aldrich) was administered to mice at 50 mg/kg.

### 2.8. Glucose Uptake Assay

The insulin resistance of high FA-induced hepatic cells was as described previously [27]. Hepa1-6 cells were seeded for 24 h. The cell culture medium was then exchanged with an FA-conditioned medium (25 mM glucose, and 0.5 mM palmitic acid and oleic acid) containing/not containing ginsenosides for 36 h. The cells were then treated with 1 nM insulin in serum-free low-glucose DMEM (5 mM glucose) for 12 h. The cell culture supernatants were collected for measurement (E1010, Applygen, Beijing, China) using a microplate reader (Infinite 50, Tecan, Mannedorf, Switzerland).

### 2.9. UPLC-MS Assay

An UPLC (Ultimate 3000 system, Thermo Scientific, Groningen, The Netherlands)-Q Exacactive-orbitrap MS system was used. A Hypersil Gold C18 column (14784, Thermo Scientific, Waltham, MA, USA) was used to determine the intracellular concentration of CK. A mobile phase consisting of acetonitrile/water was pumped through the column at 0.2 mL/min. Under these conditions, the following elution gradient was used: 50% B; 1–4 min, 50% B-95% B; 4–4.5 min, 95% B; 4.5–4.7 min, 95% B-50% B; 4.7–6 min, 50% B.

### 2.10. GR Competitor Assay

Receptor-binding assays were performed using Hepa1-6 cells pre-incubated with CK or dexamethasone (D1756, Sigma-Aldrich) for 1 h before stimulation with Fluor-Dex (D1383, Thermo Scientific). The excitation and emission wavelengths were 485 nm and 535 nm, respectively. The readings were acquired using a Spark^TM^ 10M (Tecan, Mannedorf, Switzerland).

### 2.11. Proteomics Analysis

Liver tissues were extracted in lysis buffer, and centrifuged at 10,000× *g* at 4°C for 20 min. Then, 8 M urea buffer was added and washed twice. The proteins were reduced with 10 mM DTT at 37 °C for 1 h and then alkylated by 20 mM iodoacetamide in the darkroom for 1 h. After centrifugation at 10,000× *g* for 20 min, this was changed to 50 mM NH_4_HCO_3_, and were then washed twice. Finally, proteins were resuspended in 50 mM NH_4_HCO_3_ and 1 μg trypsin at an enzyme/substrate ratio of 1:25 (*w*/*w*), and incubated at 37 °C overnight. The digestion was stopped by adding 10% TFA at a final concentration of 0.4% (*v*/*v*). The sample was desalted using a homemade C18 solid phase extraction column, centrifugally concentrated at 45 °C for 3 h, and then redissolved in 0.1% formic acid buffer. Peptides were digested with trypsin, and separated using an EASY-nLC 1200 liquid chromatographic system prior to detection on a Q-Exactive mass spectrometer equipped with a nano-electrospray ion source (Thermo Fisher Scientific, Waltham, MA, USA). Raw data from the MS were processed using the software Proteome Discoverer (Thermo Fisher Scientific, Waltham, MA, USA), and a heatmap was plotted using heatmap tools on the Genescloud platform (https://www.genescloud.cn (accessed on 22 March 2021)).

### 2.12. FA Oxidation Assay

FA’s oxidative metabolism was quantified using an Agilent Seahorse XFp Analyzer. The cells were then cultured in FA (0.5 mM) or FA plus CK (35 μM). The cells were subjected to a mitochondrial stress test by adding oligomycin (1 μM), followed by carbonyl cyanide 4-(trifluoromethoxy) phenylhydrazone (FCCP, 0.5 μM), and antimycin A/rotenone (1 mM/1 μM). The oxygen consumption rate was then measured according to the manufacturer’s instructions (Seahorse Bioscience, North Billerica, MA, USA).

### 2.13. Immunofluorescence

Hepa1-6 cells were treated with FA or FA plus CK (35 μM) for 24 h. The cells were fixed and permeabilized. For autophagy analysis, the cells were treated with WIPI2 (Abcam, 1:100) antibodies or FIP200 (CST, 1:100) antibodies for 1 h, and then treated with Alexa Fluor 568-conjugated AffiniPure goat anti-rabbit IgG (Invitrogen, A11011, 1:100) at room temperature for 1 h. For lipophagy analysis, the cells were incubated with LC3 (CST, 1:100) antibodies or LAMP1 (CST, 1:100) antibodies for 1 h. After three washes, the cells were treated with Alexa Fluor 568-conjugated AffiniPure goat anti-rabbit IgG and BODIPY 493/563 (Thermo, D3922, 1:1000) at room temperature for 1 h. For staining the GRs, the cells were treated with dexamethasone (100 nM, D1756, Sigma-Aldrich) or CK for 2 h. Further details about the methods are included in the Supporting Information (Appendix B).

### 2.14. Pull-Down Assay

CK (10 μM) and disuccinimidyl carbonate (30 μM) were dissolved in 200 μL dimethylformamide, respectively. The mixture was stirred at 50 °C for 1 h to obtain activated CK. The activated CK and beads suspension were mixed and vortexed for 1 h. Hepa1-6 cell lysates were incubated with CK plus Affigel beads, CK plus CK beads, or CK beads for overnight, respectively. The binding activity of CK and the receptors were detected using Western blot.

### 2.15. Vector Construction and Dual Luciferase Reporter Assay

To measure promoter activity, 293T cells were seeded at a density of 2 × 10^5^ cells per well into 6-well plates and cultured overnight. Then the luciferase reporters were co-transfected with 2000 ng of the promoter mixed with 50 ng of renilla (Promega, Madison, WI, USA) luciferase vector (pRL-TK) and 35 μm CK using Lipofectamine 3000 Transfection Reagent (L3000015, Thermo Fisher Scientific, Waltham, MA, USA). Further details about methods are included in the Supporting Information (Appendix B).

### 2.16. Chromatin Immunoprecipitation (ChIP)

ChIP assays were performed using a ChIP Assay kit (CST, 9003), according to the manufacturer’s instructions. In brief, Hepa1-6 cells were treated with or without CK (35 μm) for 24 h. Then, the cells were fixed in 1% formaldehyde (RHAWN, R010480) for 10 min. DNA was sheared to fragments of 150–900 bp through sonication. Lysates containing soluble chromatin were incubated and precipitated overnight with 10 μg anti-GR antibody (CST, 12041s) or normal rabbit IgG (CST, 2729). The DNA was subsequently reversed, purified, diluted, and subjected to real-time qPCR. The sequences of the qPCR primers are provided in Appendix B, Table A2.

### 2.17. Statistical Analysis

The results were expressed as the means ± SD values. Statistical analysis was performed with GraphPad Prism 8 (GraphPad Software, Inc., San Diego, CA, USA). Comparisons between the two groups were conducted with an unpaired Student’s *t* test. Data with two independent variables were tested by two-way repeated measures ANOVA with Dunnett’s post hoc test (IBM SPSS Statistics 17.0, Armonk, NY, USA). All experiments were repeated at least three times. The level of significance was defined at *, *# p* < 0.05; **, *## p* < 0.01; ***, *### p* < 0.001.

## 3. Results

### 3.1. CK Reduces Body Weight and Blood Glucose in Obese Mice

To explore whether ginsenoside CK plays a role in metabolic regulation, leptin-deficient obese mice (ob/ob) were subjected to different concentrations of CK (5, 10, 20 mg/kg/d) for 5 weeks. Compared with C57BL/6J mice, the ob/ob mice had significantly increased their levels of body weight and fasting blood glucose during the 5-week experiments (Appendix A). Notably, the body weight and fasting blood glucose significantly decreased after 20 mg/kg/d CK treatment for 5 weeks (Appendix A). In addition, the glucose tolerance test (GTT) and insulin tolerance test (ITT) were significantly ameliorated in a dose-dependent manner (Appendix A). However, the 20 mg/kg/d CK treatment did not affect food intake regardless of whether this treatment took place in the day or in the dark (Appendix A), whereas the ob/ob mice have an increased food intake in the dark, compared with the C57 mice, which is consistent with previous studies [28].

To rigorously illustrate the effect of ginsenoside CK, the weight loss drug orlistat was chosen as positive control. The body weight loss caused by 20 mg/kg/d of CK treatment is comparable to treatment with the weight loss drug orlistat. It significantly reduced the body weight and fat mass in mice (Figure 1a,b and Appendix A). During treatment with CK, the fasting blood glucose levels gradually decreased (Figure 1c), and the glucose and insulin tolerances both markedly improved (Figure 1d,e). CK also reduced the serum insulin levels (Figure 1f). Moreover, CK increased the phosphorylation levels of the insulin receptor and its substrate, indicating that CK increased insulin sensitivity in the liver (Appendix A). These results suggest that CK lowers blood glucose and increases insulin sensitivity. Thus, CK could reduce body weight and blood glucose in obese mice.

### 3.2. CK Exerts Hypolipidemic Activity in Obese Mice

To further investigate the reasons for the weight loss of CK, we analyzed the effect of CK on the metabolic activity of mice. CK treatment increased oxygen consumption (Figure 2a) and carbon dioxide production in mice (Figure 2b), and reduced respiratory exchange rate (RER), indicating that CK promoted fat consumption, whereas it did not promote carbohydrate consumption (Appendix A). As anticipated, CK increased caloric release in mice, which was consistent with the promotion of fat consumption and reduction in body weight (Figure 2c). It also reduced the weight of white adipose tissue and liver (Appendix A), the serum levels of TGs, non-esterified fatty acids (NEFAs), and total cholesterol, which indicated an overall decrease in blood lipid levels (Figure 2d–f). CK reduced the TG and NEFA levels in the liver (Appendix A), decreased the number of LDs in the liver, and the size of the adipocytes in adipose tissue, which indicates that CK affects fat deposition (Figure 2g). These results indicated that CK possesses hypolipidemic activity.

### 3.3. Ginsenoside CK Enhances Lipase Activity and Autophagy Levels to Stimulate Lipolysis

To clarify the hypolipidemic mechanism of CK, proteomics was used to screen differential proteins. Proteomic analysis was used to assess the changes in mouse livers treated with CK. Data showed that 285 proteins were identified in ob/ob versus the wild type, and 108 proteins were identified in ob/ob versus the ob/ob treated with CK, of which 60 overlapped in the wild type and CK treatment (Figure 3a). We further mapped the GO enrichment analysis, and the top 30 enriched pathways were highly clustered in metabolic pathways, especially in lipid metabolic pathways (Figure 3b). As shown in the heatmap, CK upregulated autophagy-related factors and fatty acid oxidation-related factors, and downregulated fatty acid synthesis-related factors (Figure 3c). Thus, we examined the effect of CK on autophagy and lipid metabolism. CK increased p62 degradation and LC3-I to LC3-II conversion, suggesting that CK elevated autophagy levels (Figure 3d). The AMPK/mTOR is the main signaling pathway in autophagy activation. We also detected whether they were involved in CK-induced autophagy. We found that CK activated the AMPK signaling pathway, rather than the mTOR signaling pathway in the liver (Figure 3d). Treatment with CK significantly increased the mRNA and protein expression of PPARα, ACOX1, and CPT1α, indicating that it promotes fat catabolism (Figure 3e,f). In addition, CK enhanced the expression of lipases (ATGL and HSL), and stimulated the cAMP/PKA/HSL signaling pathway to promote lipolysis (Figure 3g). CK also inhibited the expression of factors involved in lipogenesis, indicating negatively regulates fat synthesis (Figure 3h,i). Together, CK enhanced autophagy and lipolysis in the liver of obese mice.

### 3.4. CK Activates Autophagy by Activating AMPK/ULK1 Pathway

To identify the role of CK in regulating autophagy, we studied the molecular mechanism of CK to activate autophagy. Using GFP-LC3-transfected HeLa cells, we tested the autophagy induction ability of CK. The number of autophagosomes with CK treatment was increased in a dose-dependent manner. The EC50 values in cells treated with CK was 35 μM (Appendix A). After 15 min of treatment with 35 μM of ginsenoside CK, autophagosomes were observed to form. The number of autophagosomes increased with time, indicating that CK rapidly activated autophagy (Figure 4a). The inhibitor 3-MA impaired CK-induced autophagy at an early stage (15 min), while BafA1 impaired CK-induced autophagosome degradation during a later stage (3 h), suggesting that CK activates autophagy by inducing the appearance of the phagophore (Figure 4a). Ginsenoside CK exhibited the highest potential for activating autophagy not only in HeLa cells, but also in HepG2 and A549 cells (Figure 4b). After the addition of the autophagy inhibitors 3-methyladenine (3-MA, which blocks class III phosphatidylinositol 3-kinase activity) or bafilomycin A1 (BafA1, which blocks autophagosome and lysosome fusion), the degradation of p62 and the conversion of LC3-I to LC3-II were significantly suppressed compared to the control (Figure 4b and Appendix A). The results further indicate that ginsenoside CK activates autophagy. The formation of the phagophore requires FIP200, while the maturation of autophagosomes requires WIPI2 [29,30]. We found that LC3 puncta produced by CK co-localized with FIP200 puncta at 15 min, whereas they co-localized with WIPI2 puncta at 30 min. Therefore, it is more likely that CK initiated autophagy from phagophore formation (Figure 4c).

We also analyzed the signaling pathways involved in CK-induced autophagy. The ULK1 as a downstream signaling molecule of AMPK and mTOR has two phosphorylation sites. Activated AMPK can phosphorylate ULK1 at S555, while activated mTOR phosphorylates ULK1 at S757 [31]. The results showed that CK treatment activated AMPK-induced phosphorylation at S555 of ULK1, but did not affect the phosphorylation of ULK1 at S757, indicating that CK activated AMPK (Figure 4d). In addition, p70S6K and 4EBP1 are both downstream elements of mTOR. After treatment with CK, there is no effect on the activation of mTOR and its downstream molecules, indicating that CK does not affect mTOR. Therefore, CK activates autophagy in an AMPK/ULK1-dependent and mTOR-independent manner.

We attempted to confirm whether CK induces autophagy in vivo. A CK injection at a dose of 20 mg/kg/d markedly increased p62 degradation, while administration of the inhibitor chloroquine, which elevates the lysosomal pH to inhibit cargo degradation, inhibited CK-induced p62 degradation (Figure 4e). Based on these finding, CK activates autophagy by activating the AMPK/ULK1 pathway.

### 3.5. Ginsenoside CK-Stimulated Lipophagy and Lipase Activity Are Independent

Specific lipophagy is associated with the degradation of fats. We attempted to determine if autophagy results from CK-induced lipophagy, and if the relationship between autophagy and lipolysis induced by CK. We treated Hepa1-6 hepatocytes with high concentrations of FA to mimic the conditions observed in the liver of obese mice. FA reduced the uptake of glucose in cells, resulting from an accumulation of LDs (Figure 5a,b). However, after CK was added, the glucose absorption of cells increased (Figure 5a), and the number of LDs decreased (Figure 5b), which indicates that CK improved insulin resistance in hepatocytes caused by FA. This was consistent with the results observed in vivo. The autophagy inhibitor BafA1 partially attenuated the effect of CK on insulin resistance in cells, suggesting that autophagy is one of the mechanisms involved (Figure 5a,b). To further confirm the effect of autophagy, autophagy was blocked by treatment with siRNA of ATG7. It was observed that the phosphorylation of the insulin receptor was inhibited, indicating that CK improved insulin resistance through autophagy (Figure 5c). In obese mice, we found that CK enhanced lipolysis in the liver. In Hepa1-6 cells, we also observed that CK enhanced beta oxidation (Figure 5d). CK increased the co-localization of LDs (BIODIPY) and autophagosomes (LC3), as well as LDs (BIODIPY) and lysosomes (LAMP1), indicating that CK specifically activated lipophagy to reduce the number of LDs (Figure 5e). However, we observed that CK enhanced the expression of lipolytic factors, while the ATG7 siRNA did not affect the expression of lipase induced by CK, indicating that autophagy and lipase expression are independent (Figure 5f). Therefore, CK can enhance lipolysis by two independent processes, including the induction of specific lipophagy and enhancement of lipase activity.

### 3.6. Ginsenoside CK Acts on the Glucocorticoid Receptor (GR) to Promote Lipolysis

Now that CK was observed to activate lipophagy and enhance lipases expressions in hepatocytes, we needed to reveal the target of CK. Quantitative analysis using UPLC-MS showed that CK entered the cells and reached a maximum concentration of 109.3 pmol/g at 48 h (Figure 6a). Since CK has a steroid-like structure, we proposed that CK binds to G protein-coupled receptors involved in lipid metabolism. The glucocorticoid receptor (GR) and the mineralocorticoid receptor (MR) could bind to glucocorticoids to play an important role in lipid metabolism [32]. Bile acid receptor (TGR5) or adrenergic receptor (β3-AR) were known to activate adenylate cyclase (AC) to increase intracellular cAMP levels [33,34]. The pull-down assay confirmed that CK could strongly bind to GR, but not TGR5, MR, or β3-AR (Figure 6b). Furthermore, CK was observed to compete with fluorescently labeled dexamethasone for GR binding (Figure 6c). CK also increased the nuclear translocation of the GR, as well as of dexamethasone (Figure 6d). These results indicate that CK enters the cell and binds to the GR. The mRNA level of ATGL was upregulated by CK treatment, but RU486 as a GR inhibitor suppressed ATGL expression. We speculated that GR could regulate ATGL transcription by binding to its promoter (Figure 6e). To identify the sites that are critical for the regulation of ATGL transcription, we amplified the promoter sequence from −1210 to +50, and divided it into four fragments that were individually inserted into reporter constructs for the dual luciferase assay. The fragment from −430 to +50 was associated with the highest luciferase activity (Figure 6f). Furthermore, GR markedly increased ATGL promoter activity through the −430 to −310 region (Figure 6g). These results indicated that CK promoted lipolysis through GR-induced ATGL transcription enhancement. Treatment with GR siRNA impaired the increase in insulin sensitivity caused by CK (Figure 6h), inhibited the activation of autophagy and AMPK/ULK1 signaling transduction (Figure 6i), as well as the enhancement of lipase expression (Figure 6j). Treatment with the antagonist RU486 consistently reduced the increase in lipolytic factor expression caused by CK (Figure 6k). Together, ginsenoside CK acted on GR to promote lipophagy and lipase activity (Figure 6l).

## 4. Discussion

Autophagy is necessary for maintaining metabolic homeostasis in human body, and certain autophagy activators have been used to treat metabolic diseases [35]. Our previous studies found that ginsenoside Rg2 can activate autophagy to improve insulin resistance [22].

Ginsenoside is a natural molecule with a steroid-like structure, possessing hundreds of monomers due to its aglycone and sugar moiety. Here, we observed that the reduction in body weight, fat mass, blood glucose levels, and lipid levels in obese mice upon treatment with CK appears to be mediated via autophagy-dependent mechanisms (Figure 1, Figure 2 and Figure 3). CK activates autophagy in cells and in mice, and is able to induce autophagy via phagophore formation (Figure 4). This effect of CK is not only found in hepatocytes, but also in HeLa cells, HepG2 cells, and A549 cells (Figure 4), which indicates that CK induced autophagy universally. Moreover, CK activates AMPK/ULK1, and specifically induces lipophagy. This is the first demonstration of the regulation of lipophagy by CK (Figure 5). Therapeutic strategies to increase autophagic function may therefore provide a new approach to prevent the metabolic syndrome and its associated pathologies. We also found that CK and dexamethasone competitively bind to GR, indicating that CK and dexamethasone may have similar structures. Recently, dexamethasone has been proved to induce brown adipose tissue whitening and fat mass gain by autophagy activation [36]. However, CK promotes weight loss, indicating that CK is not a substitute for glucocorticoid (GC).

Lipid accumulation in white adipose tissue (WAT) is mainly regulated by hormones and adipocytokines, including insulin, adiponectin, leptin, tumor necrosis factor α (TNFα), and interleukins (ILs) [37]. These adipokines are described to be involved in mediating insulin resistance in peripheral tissues, as well as to increase the risk of obesity [38]. We observed that CK-treated mice have significantly lower WAT mass and significantly smaller adipocytes compared to ob/ob mice (Figure 2g and Appendix A). Thus, CK may affect obesity-related adipocytokines. Further studies will be linked to the secretory function of CK-mediated adipocytes to obesity.

It has been reported that treatment with rapamycin, an activator of autophagy, significantly decreased LD number and TGs, and increased LD/LAMP1 co-localization [19]. As for our results, CK also increased LD/LC3 and LD/LAMP1 co-localization in FA-treated cells (Figure 5e). However, treatment with the autophagy inhibitor BafA1 and the siRNA-mediated knockdown of Atg7 did not completely inhibit CK-mediated fat loss (Figure 5f). The classically physiological regulation of cytosolic lipolysis is modulated by catecholamines, hormones, and growth factors, which commonly act on cyclic adenosine monophosphate (cAMP)–protein kinase A (PKA), signaling to control lipase activity, for example the phosphorylates activity of HSL [39]. Therefore, we also tested whether CK enhanced lipolysis via activating cAMP/PKA/HSL pathway (Figure 3g). Based on these findings, we propose that CK reduces fat deposition by inducing lipophagy and lipolysis. The results of proteomics also further verify that CK does regulate autophagy-related proteins and fatty acid metabolism-related proteins (Figure 3a–c). The most distinct differences in expression of autophagy regulatory proteins are EVA1A (eva-1 homolog A) and USP10 (ubiquitin carboxyl-terminal hydrolase 10). EVA1A could interact with ATG16L1 to locate in the autophagosome membrane, thereby promoting the formation of the autophagosomes [40]. USP10 is the important member of the deubiquitinating enzymes family, which mediates the de-ubiquitination of Beclin1 to stabilize the PIK3C3/VPS34 complex [41]. The mechanism by which CK regulates autophagy through these two target proteins is still in progress.

In general, autophagosomes can only engulf smaller LDs and transport them to lysosomes for degradation [42,43]. Larger LDs (diameter > 10 µm) also depend on cytosolic lipases to degrade them into FA. Here, we observed that the suppression of autophagy does not affect the enhancement of lipase activities by CK, indicating that both lipophagy and lipases contribute to the role of CK in lipolysis (Figure 5). We intend to focus on the molecular mechanisms underlying the crosstalk between lipophagy and lipases in future research.

Adreno–cortical hormones and catecholamines are important regulators of lipid metabolism [44]. There are two primary nuclear receptors, GR and MR, to bind to GC. Low-dose chronic treatment of GC preferentially activates the lipogenic actions of adipose MR. whereas high-dose acute treatment is needed to activate GR-mediated lipolysis [45]. GC binds to GR, causing a dissociation of Hsp90, a translocation to the nucleus, and binding to GC response elements (GRE) in promoters of lipolysis genes [46]. Once catecholamines bind to the ligands to G protein-coupled receptors, the activation of adenylyl cyclase follows, leading to an increase in cAMP content, which consequently leads to the activation of the PKA/HSL signaling pathway for the promotion of lipolysis [47]. Ginsenosides are structurally similar to steroid hormones. Ginsenoside Rg1 has been reported to serve as a functional ligand of glucocorticoid receptor [48,49]. Here, we observed that, compared to TGR5, MR, and β3-AR, CK mainly binds to GR (Figure 6). Furthermore, CK promotes the binding of GR at the lipase ATGL promoter (Figure 6). Consistent with our work, Desarzens et al. reported that ATGL expression and lipolysis were decreased in mice lacking GR in adipocytes, protecting mice from dexamethasone-induced metabolic dysfunction [50]. However, the chances of CK binding to other receptors cannot be ruled out, and future studies are necessary to assess whether other potential G protein-coupled receptors also contribute to the therapeutic effects of CK.

## 5. Conclusions

Our data demonstrate that CK enhanced lipophagy and ATGL activity, reduced fat accumulation, improved sensitivity to insulin in the liver, and reduced blood glucose and lipid levels, along with body weight. Therefore, CK visibly lowers fat and weight. Our research data will provide a candidate molecule for improving obesity, diabetes, and other lipid metabolism related diseases therapy.

## Figures and Tables

**Figure 1 pharmaceutics-14-01192-f001:**
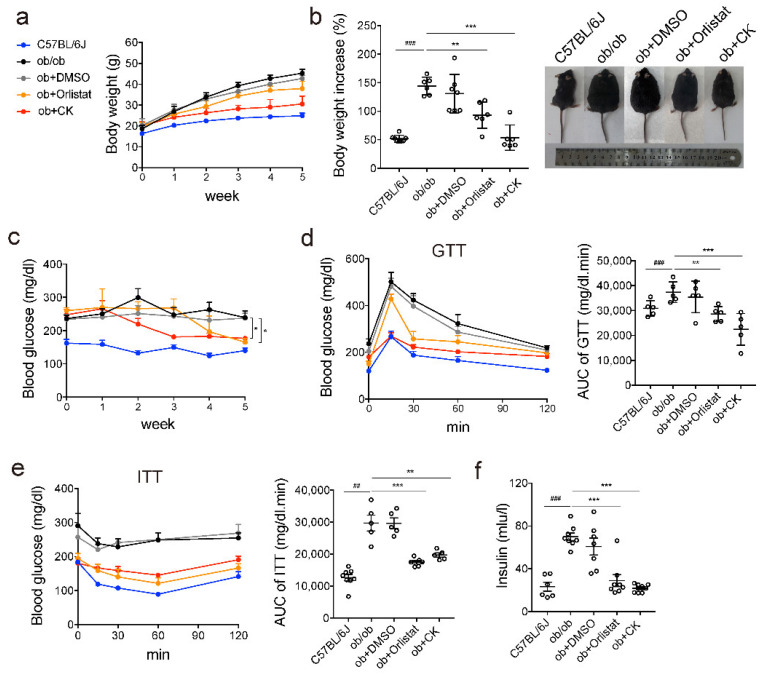
CK treatment ameliorates adiposity and blood glucose in ob/ob mice. Obese mice were orally administered 150 mg/kg of orlistat, or injected with the vehicle (DMSO) or 20 mg/kg of the indicated ginsenosides for 5 weeks (*n* = 5–8). (**a**) Body weight. (**b**) Body weight increase rate and representative photos of mice. (**c**) Fasting blood glucose levels. (**d**) Glucose tolerance test and area under the receiver operating characteristic curve (AUC). (**e**) Insulin tolerance test and AUC. (**f**) Serum insulin content. Statistical data represent the comparison of each value in the compound-treated groups to the corresponding value in the ob/ob groups. Results are presented as mean ± SD. *##*, *p* < 0.01; *###*, *p* < 0.001; *, *p* < 0.05; **, *p* < 0.01; ***, *p* < 0.001.

**Figure 2 pharmaceutics-14-01192-f002:**
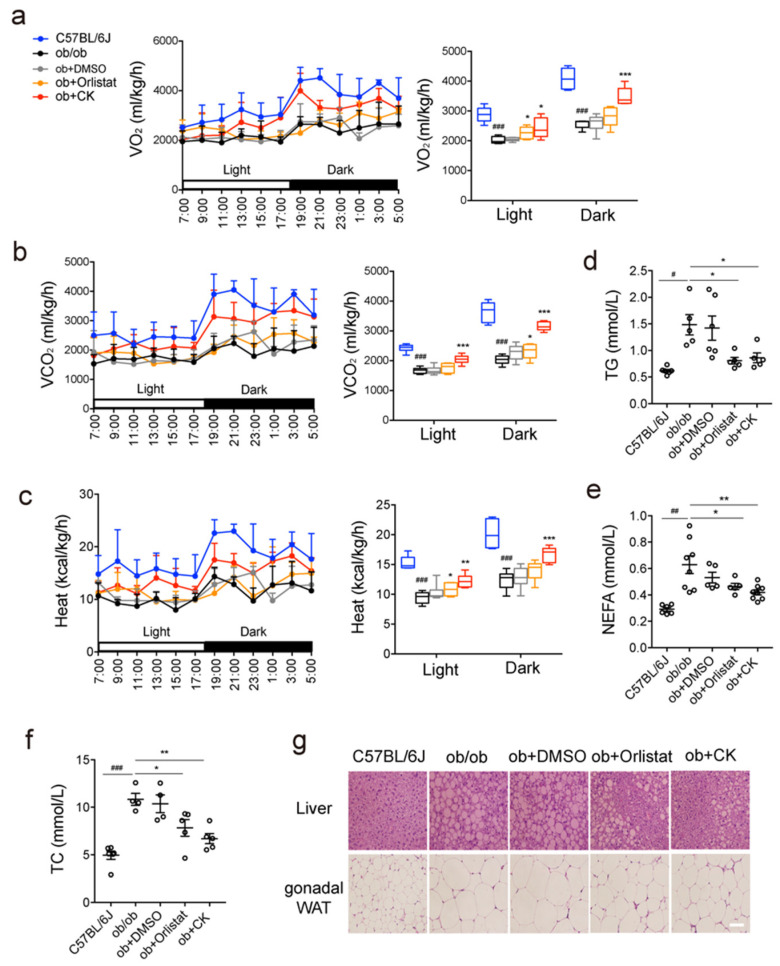
Metabolic profiles and lipid levels of obese mice after CK treatment. Obese mice were orally administered 150 mg/kg of orlistat, or were injected with the vehicle (DMSO) or 20 mg/kg of the indicated ginsenosides for 5 weeks (*n* = 4–10). (**a**) Oxygen consumption. (**b**) Carbon dioxide production. (**c**) Heat production. (**d**) Serum triglyceride levels. (**e**) Serum non-esterified fatty acid levels. (**f**) Serum total cholesterol levels. (**g**) Representative images of fat deposition in liver and white adipose tissue by hematoxylin eosin staining. Scale bar, 100 μm. Statistical data represent the comparison of each value in the compound-treated groups to the corresponding value in the ob/ob groups. Results are presented as mean ± SD. *#*, *p* < 0.05; *##*, *p* < 0.01; *###*, *p* < 0.001; *, *p* < 0.05; **, *p* < 0.01; ***, *p* < 0.001.

**Figure 3 pharmaceutics-14-01192-f003:**
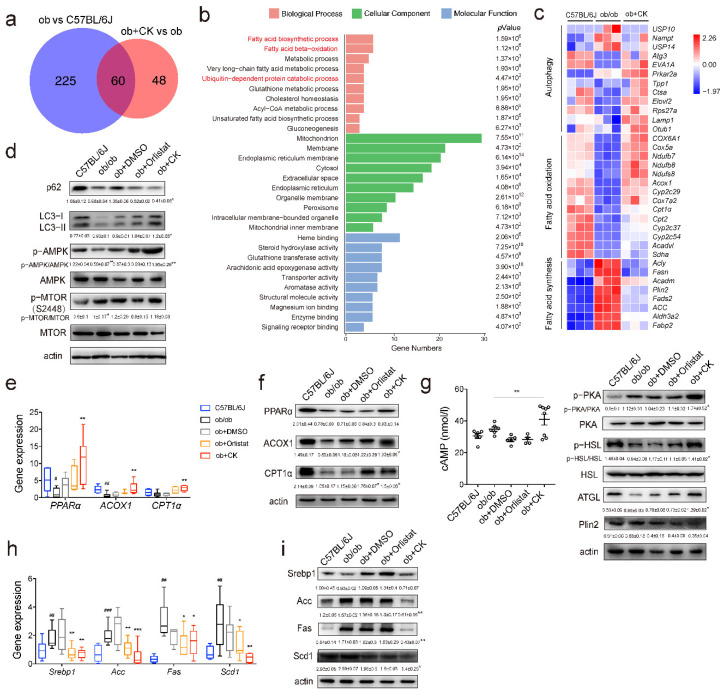
CK suppresses hepatic fat deposit through the enhancement of lipase expression and autophagy. (**a**) Venn diagrams showing the number of unique or shared genes between ob vs. C57BL/6J and ob + CK vs. ob datasets. The common part (60 genes) likely reflects CK effect. (**b**) GO enrichment analysis predicted the top 30 enriched pathways of CK-regulated genes. Rich factor is the ratio of the differentially expressed gene number to the total gene number in a certain pathway. *Q*-value is corrected *p*-value ranging from 0–1. (**c**) Heat map of changes in protein levels of autophagy-related and fatty acid catabolism-related factors. (**d**) Autophagy signaling pathways analyzed using Western blotting of liver tissues (*n* = 3). (**e**) Gene expression of lipolysis factors assessed using qRT-PCR (*n* = 4–10). (**f**) Protein expression of lipolysis factors assessed using Western blotting (*n* = 3). (**g**) cAMP/PKA/HSL signaling pathways were analyzed using Western blotting of liver tissues (*n* = 3). (**h**) Gene expression of lipogenesis factors assessed using qRT-PCR (*n* = 4–10). (**i**) Protein expression of lipogenesis factors assessed using Western blotting (*n* = 3). Statistical data represent the comparison of each value in the compound-treated groups to the corresponding value in the ob/ob groups. Results are presented as mean ± SD. *#*, *p* < 0.05; *##*, *p* < 0.01; *###*, *p* < 0.001; *, *p* < 0.05; **, *p* < 0.01; ***, *p* < 0.001.

**Figure 4 pharmaceutics-14-01192-f004:**
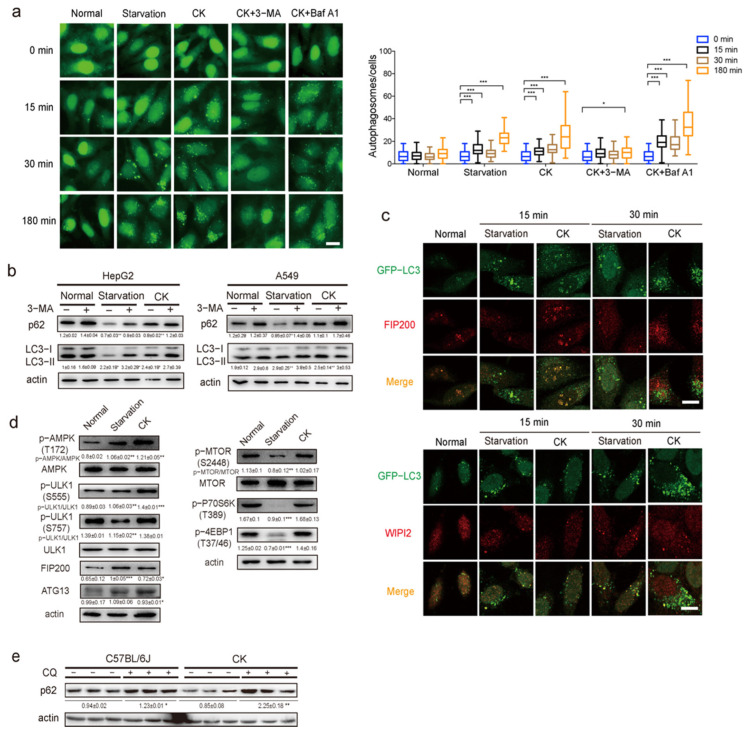
Ginsenoside CK induces autophagy in vitro and in vivo. (**a**) Representative images and quantification of GFP-LC3 puncta in HeLa cells stably expressing GFP-LC3 cultured in normal or starvation medium, or treated with CK in normal medium, in the presence or absence of lysosomal inhibitors 3-MA or BafA1 for the indicated time period (*n* = 100). Statistical data represent the comparison of each value to the corresponding value under normal conditions. Scale bar, 20 μm. (**b**) Western blotting detection of p62 and LC3 in HepG2 cells (left) or A549 cells (right) cultured in normal or starvation medium, or treated with CK in normal medium, in the presence or absence of lysosomal inhibitors 3-MA for 3 h (*n* = 3). (**c**) Representative images of co-localization of LC3 (green) and FIP200 (red, upper) or WIPI2 (red, lower) in normal or starvation medium, or when treated with CK in normal medium for the indicated time. Scale bar, 10 μm. (**d**) AMPK and mTOR signaling pathways were analyzed by western blotting in Hepa1-6 cells cultured in normal or starvation medium, or treated with CK in normal medium for 3 h (*n* = 3). (**e**) Western blotting detection of p62 in liver of mice injected with CK, in the presence or absence of the lysosomal inhibitor chloroquine (*n* = 3). Results are presented as mean ± SD. *, *p* < 0.05; **, *p* < 0.01; ***, *p* < 0.001.

**Figure 5 pharmaceutics-14-01192-f005:**
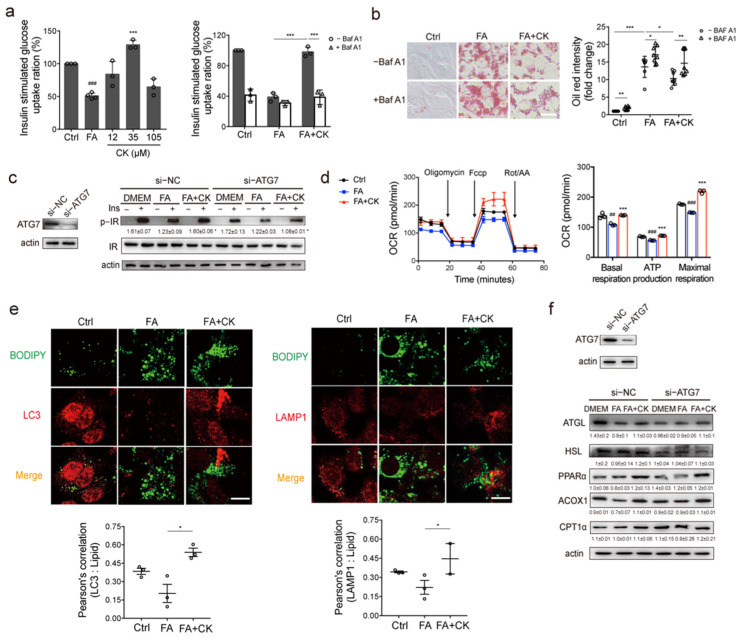
CK activates lipophagy and lipase activity. Hepa1-6 cells were cultured in normal medium, 0.5 mM free fatty acid (FA) medium, or with indicated ginsenosides in FA medium. (**a**) Insulin-stimulated glucose uptake of the cells cultured in FA medium with the presence or absence of the lysosomal inhibitor BafA1 (*n* = 3). (**b**) Representative images of the lipid droplets in cells visualized by oil red staining, in the presence or absence of the lysosomal inhibitor BafA1 (left), and the quantification of lipid droplets (right) (*n* = 3). Scale bar, 100 μm. (**c**) The insulin receptor signaling pathways were analyzed using western blotting (*n* = 3). The cells were transfected with control (NC) or ATG7 siRNAs 36 h prior to FA or CK treatment. (**d**) Fatty acid oxidation was analyzed in cells treated with FA or CK using an Agilent Seahorse XFp Analyzer (*n* = 3). (**e**) Representative images of the co-localization of lipid droplets (green) and autophagosomes (red), and co-localization of lipid droplets (green) and lysosomes (red). Scale bar, 10 μm. (**f**) Protein expression of lipolysis factors were analyzed using western blotting (*n* = 3). The cells were transfected with control (NC) or ATG7 siRNAs 36 h prior to FA or CK treatment. Results are presented as mean ± SD. *##*, *p* < 0.01; *###*, *p* < 0.001; *, *p* < 0.05; **, *p* < 0.01; ***, *p* < 0.001.

**Figure 6 pharmaceutics-14-01192-f006:**
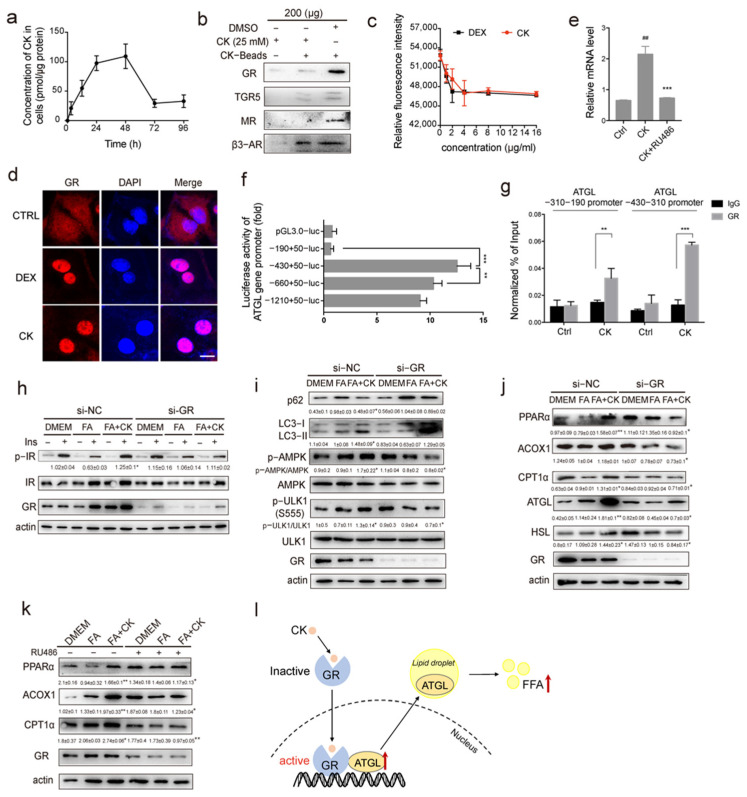
CK enhances lipolysis and autophagy by binding to ATGL promoter through the glucocorticoid receptor (GR). (**a**) Intracellular transport of CK in hepatocytes was analyzed using UPLC-MS (*n* = 3). (**b**) The binding ability of CK to G-protein-coupled receptors were evaluated in the pull-down assay. (**c**) The competitive combination of CK and dexamethasone with GR was analyzed in the fluorescence quantitative assay (*n* = 3). (**d**) Representative images of the nuclear translocation of GR in hepatocytes treated with dexamethasone or CK. Scale bar, 10 μm. (**e**) qPCR analysis of ATGL mRNA levels treated with CK or RU486 (*n* = 4–6). (**f**) ATGL promoter vector dual luciferase statistical results (*n* = 3–8). (**g**) ChIP-qPCR analysis showing hepatocytes exposed to CK for 24 h. IgG was used as a negative control (*n* = 4). (**h**) The insulin receptor signaling pathways were analyzed using western blotting (*n* = 3). Cells were transfected with the control (NC) or GR siRNAs 36 h prior to FA or CK treatment. (**i**) Autophagy signaling pathways were analyzed using western blotting (*n* = 3). The cells were transfected with the control (NC) or GR siRNAs 36 h prior to FA or CK treatment. (**j**) The expression of lipolysis factors and lipase expression were analyzed using western blotting (*n* = 3). The cells were transfected with the control (NC) or GR siRNAs 36 h prior to FA or CK treatment. (**k**) The expression of lipolysis factors was analyzed using western blotting in the presence or absence of the GR inhibitor RU486 (*n* = 3). (**l**) Schematic diagram illustrating the model that CK drives the nucleocytoplasmic transport of GR, subsequently increasing the transcriptional activity of the ATGL gene, resulting in enhanced protein expression and lipolysis activity of ATGL. Results are presented as mean ± SD. *##*, *p* < 0.01; *, *p* < 0.05; **, *p* < 0.01; ***, *p* < 0.001.

## Data Availability

The data presented in this study are available in the article.

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
