# Peer review of "Ginsenoside Compound K Protects against Obesity through Pharmacological Targeting of Glucocorticoid Receptor to Activate Lipophagy and Lipid Metabolism"

_pharmaceutics, 2022, doi:10.3390/pharmaceutics14061192_

Round 1

Reviewer 1 Report

Dear the Editor

The authors demonstrated the efficacy of Ginsenoside compound K (CK) on obesity using ob/ob mice model. At 20mg/kg, the increase of body weight was almost comparable to the control mice. They further hypothesized that whether CK might activate lipophagy through glucocorticoid receptor-mediated mechanism. This was supprted by subsequent biochemical experiments. Current Figures are too small to see the data. Thus, some data and experimental detail which is associated with this manuscript, but not essential for displaying items, are to be placed as Supplementary Materials.

Author Response

The authors demonstrated the efficacy of Ginsenoside compound K (CK) on obesity using ob/ob mice model. At 20mg/kg, the increase of body weight was almost comparable to the control mice. They further hypothesized that whether CK might activate lipophagy through glucocorticoid receptor-mediated mechanism. This was supported by subsequent biochemical experiments. Current Figures are too small to see the data. Thus, some data and experimental detail which is associated with this manuscript, but not essential for displaying items, are to be placed as Supplementary Materials.

Responses to the Comments: We are grateful for your suggestions. We have placed some data and experimental detail as Appendix A and supplementary figures, please see the revised manuscript and supplementary files.

Reviewer 2 Report

In this study, the authors showed that ginsenoside compound K (CK) suppresses obesity by targeting glucocorticoid receptors and activating lipocytosis and lipid metabolism. Based on this result, this paper suggests that CK may be a novel GR agonist for the treatment of obesity and has academic value. Please refer to the minor comments below.

1) Reading from line 302, I think the CK processing concentration is 20 mg / kg / d. In order to make the reader's understanding accurate, I think it is good to show the numerical values in Fig. 1-f.

2) Fig. 1f, which is described in line 322, seems to be correct in Fig. 1g. please confirm.

3) I think the letters in the figures are small. Why don't you write it a little bigger?

Author Response

In this study, the authors showed that ginsenoside compound K (CK) suppresses obesity by targeting glucocorticoid receptors and activating lipocytosis and lipid metabolism. Based on this result, this paper suggests that CK may be a novel GR agonist for the treatment of obesity and has academic value. Please refer to the minor comments below.

 1) Reading from line 302, I think the CK processing concentration is 20 mg/kg/d. In order to make the reader's understanding accurate, I think it is good to show the numerical values in Fig. 1-f.

Responses to the Comments: We are grateful for your suggestions and have added the CK processing concentration in Figure S1f, please see the revised manuscript.

2) Fig. 1f, which is described in line 322, seems to be correct in Fig. 1g. please confirm.

Responses to the Comments: We are grateful for your suggestions and have corrected it.

3) I think the letters in the figures are small. Why don't you write it a little bigger?

Responses to the Comments: We are grateful for your suggestions. In order to be clear, we have modified the word and image size in figures, please see the revised manuscript.

Reviewer 3 Report

Dear authors, 

Congratulations on your hard work. Osesity is a serious problem, so new strategies to induce lyposis are very important.

Only a single remark I will do. 

Please specify into materials and methods, the  diet of  obese mouse/versus the control group.  

Author Response

Please specify into materials and methods, the diet of obese mouse/versus the control group.

Responses to the Comments: We are grateful for your suggestions. The 6-weeks-old male C57BL/6J and ob/ob mice used in our experiment were maintained with same chow diet (60% cereals, 33% protein and 3% oil). We have added this information in section 2.3, please see the revised manuscript.

Reviewer 4 Report

The paper entitled “Ginsenoside Compound K protects against obesity by pharmacological targeting of glucocorticoid receptor to activate lipophagy and lipid metabolism” includes potentially relevant data for the scientific purpose as well as for the pharmacology of chosen metabolic disorders – including obesity and/or diabetes type 2. Authors of this publication – among others – tend to suggest that Ginsenoside Compound K enhanced lipophagy and ATGL activity, reduced fat accumulation, improved sensitivity to insulin in liver, and reduced blood glucose and lipid levels, along with body weight.

Remarks:

  1. The publication lacks a clearly defined aim of the undertaken scientific investigations - what the Authors wanted to achieve?
  2. Chapter conclusions summarize the results. Mentioned chapter need to be remodeled.
  3. Both in the introduction chapter and in the discussion chapter no reference to the influence of ginsenoside on the secretory function of adipocytes.
  4. Did Authors observed the trends of glycaemia after the implementation of chosen active pharmaceutical substance of natural origin?
  5. There is no information on the standardization of the animal model, the strengths and weaknesses of the animal model (implemented in the study) have not been determined, no information on the weight of the animals is available.
  6. Figure 1 is illegible. Too many partial figures gathered in one place. Must be modified.
  7. The authors should add an explanation of the statistical methods used - otherwise it is difficult - to objectively evaluate the obtained research results.

Author Response

1. The publication lacks a clearly defined aim of the undertaken scientific investigations - what the Authors wanted to achieve?

Responses to the Comments: We are grateful for your suggestions. Lipid-lowering drugs have been widely used to treat hyperlipidemia, but these drugs are all single targeted drugs with some limitations. Therefore, we aimed to find the safe and efficient lipid-lowering drugs for improving obesity, diabetes and other lipid metabolism related diseases therapy. We have added the information required, please see chapter conclusions.

2. Chapter conclusions summarize the results. Mentioned chapter need to be remodeled.

Responses to the Comments: We are grateful for your suggestions. We have enriched the discussion and added the conclusions, please see the revised manuscript.

3. Both in the introduction chapter and in the discussion chapter no reference to the influence of ginsenoside on the secretory function of adipocytes.

Responses to the Comments: We are grateful for your suggestions. We have added the influence of ginsenoside on the secretory function of adipocytes in the introduction and discussion, please see the revised manuscript.

4. Did Authors observed the trends of glycaemia after the implementation of chosen active pharmaceutical substance of natural origin?

Responses to the Comments: We are grateful for your suggestions. We have showed the fasting blood glucose levels in Figure 1c, which showed significantly decreased in ob/ob mice with 20 mg/kg/d of CK treatment, please see the revised manuscript.

Moreover, we have examined the effect of CK on blood glucose in C57BL/6J mice before, which are not shown in manuscript. The results indicated that intraperitoneal injection of CK at a dose of 20 mg/kg/d for 4 weeks had no effect on blood glucose, indicating that ginsenosides CK have no evident side effects on normal mice.

5. There is no information on the standardization of the animal model, the strengths and weaknesses of the animal model (implemented in the study) have not been determined, no information on the weight of the animals is available.

Responses to the Comments: We are grateful for your suggestions. In fact, the ob/ob mice with a genetic defect causing obesity and type II diabetes has been known since 1950. The ob gene product, called leptin, which is synthesised and secreted by adipocytes, is lacking in homozygote ob/ob mice, causing a severe obesity in mice and humans. Compared with C57BL/6J mice, the ob/ob obese model mice is characterized by hyperphagia, obesity, lower energy expenditure and activity, hyperglycemia, and hyperinsulinemia (Biomed Pharmacother. 1997, 51(8), 318-23. doi: 10.1016/S0753-3322(97)88048-1; Adv. Sci. 2021, 8, 2004032. doi: 10.1002/advs.202004032). We selected ob/ob mice which is consistent with the law of human obesity. Consistence with these reports, the ob/ob mice showed higher level of body weight, fasting blood glucose and food intake with impaired glucose tolerance (Figure 1a-f). We have added the information required, please see section 2.3 and 3.1.

6. Figure 1 is illegible. Too many partial figures gathered in one place. Must be modified.

Responses to the Comments: We are grateful for your suggestions. In order to be clear, we have modified the Figure S1, please see the supplementary files.

7. The authors should add an explanation of the statistical methods used - otherwise it is difficult - to objectively evaluate the obtained research results.

Responses to the Comments: We are grateful for your suggestions and have revised the statistical analysis.

Round 2

Reviewer 4 Report

The Authors of the publication "Ginsenoside Compound K protects against obesity by pharmacological targeting of glucocorticoid receptor to activate lipophagy and lipid metabolism" modified the manuscript with reference to the reviewer's comments.

The article (pharmaceutics-1686927) can be published.